# Public-Key Cryptosystems and Bounded Distance Decoding of Linear Codes

**DOI:** 10.3390/e24040498

**Published:** 2022-04-01

**Authors:** Selda Çalkavur

**Affiliations:** Math Department, Faculty of Arts and Science, Kocaeli University, Kocaeli 41380, Turkey; selda.calkavur@kocaeli.edu.tr

**Keywords:** public-key cryptosystem, error correcting code, bounded distance decoding

## Abstract

Error-correcting codes form an important topic in information theory. They are used to correct errors that occur during transmission on a noisy channel. An important method for correcting errors is bounded distance decoding. The public-key cryptosystem is a cryptographic protocol that has two different keys. One of them is a public-key that can be known by everyone, and the other is the private-key only known to the user of the system. The data encrypted with the public-key of a given user can only be decrypted by this user with his or her private-key. In this paper, we propose a public-key cryptosystem based on the error-correcting codes. The decryption is performed by using the bounded distance decoding of the code. For a given code length, dimension, and error-correcting capacity, the new system allows dealing with larger plaintext than other code based public-key cryptosystems.

## 1. Introduction

Public-key cryptosystems or asymmetric cryptosystems have been a subject of study since 1976. These systems consider two different keys, which are called public-key and private-key. These keys are not completely independent of each other. There must be a mathematical relationship as factoring, discrete logarithm, etc. [1,2]. The public-key cryptosystem was first introduced in 1976 by Diffie and Hellman [3]. Rivest, Shamir and Adleman’s paper, known as the RSA cryptosystem [4], also present a public-key cryptosystem. The RSA cryptosystem was based on the factorization integers [5]. Merkle and Hellman [6] suggested a cryptosystem based on the difficulty of the integer packing “knapsack” problem.

The first public-key cryptosystem based on the error-correcting codes was presented by R. J. McEliece in 1978 [7]. He has employed error correcting codes, in particular binary Goppa codes, with a known decoding algorithm to construct the system. The generator matrix *G* plays an important role. The most important property of McEliece’s cryptosystem is its large key size. Niederreiter suggested another code-based public-key cryptosystem that is based on the syndrome decoding of linear codes [8]. This system is used for the parity-check matrix *H* of a linear code. Thus, it is also the dual version of McEliece’s cryptosystem. If it is used with exactly the same parameters [9], McEliece’s cryptosystem and Niederreiter’s cryptosystem offer an equivalent security. Li et al. [10] proposed new classes of trapdoor functions to solve the bounded distance decoding problem in lattices. Moreover, a lot of cryptosystems have been presented by using linear codes after McEliece’s and Niederreiter’s schemes. The use of subcodes of generalized Reed–Solomon codes was introduced by Berger and Loidreau [11]. Berlekamp et al. [12] studied the complexity of the decoding of arbitrary linear codes. Krouk [13] proposed a different class of public-key cryptosystems. Sidelnikov [14] introduced the use of Reed–Muller codes for cryptosystems. Berger et al. [15] and Misoczki-Barreto [16] proposed using quasi-cyclic and quasi-dyadic codes to shorten the McEliece key. The original parameters of the McEliece cryptosystem have been broken [17], but the general system is still considered safe.

In this study, we propose a public-key cryptosystem based on the error-correcting codes using a known bounded distance decoding method. We present the encryption and decryption algorithms by inspiring both McEliece’s and Niederreiter’s cryptosystems.

McEliece’s system has been constructed based on linear codes over $F_{2}$.Both Niederreiter’s and our system have been constructed based on linear codes over $F_{q}$.However, in our cryptosystem, since it is easier to generate the pieces of keys, the encryption, decryption, and key generation are more effective than Niederreiter’s cryptosystem.Another difference of our system from Niederreiter’s is the use of the bounded distance decoding method, which corrects errors and guarantees unique decoding.It is impossible to find the private-key with public-key by an attacker in our public-key cryptosystem.Similarly, even if an enemy knows the public-key and ciphertext, he/she cannot calculate the plaintext.

These conditions ensure the new system is safe. Moreover, we consider some possible attacks in this paper. So, we analyze its security and performance, and we calculate some important parameters for our cryptosystem. When we compared it with McEliece’s and Niederreiter’s cryptosystems, we can say that our system performs better as regards encryption speed.

The rest of the paper is organized as follows. The next section gives the necessary background on coding theory and cryptography. Section 3 introduces the new public-key cryptosystem. Section 4 analyzes its security and examines some possible attacks. Section 5 compares it to the other code-based public-key cryptosystems. Section 6 concludes the paper.

## 2. Preliminaries

In this section, we remind of some important topics [18,19] that are necessary for the paper.

### 2.1. Linear Codes

**Definition** **1**(Linear Code)**.**
*A linear code C of length n and dimension k is a subspace of ${\left(F_{q}\right)}^{n}$, where $F_{q}$ is the finite field with q elements, q is a prime power, and k and n are positive integers such that k≤n. It is denoted by an [n,k]-code. The error-correcting capacity of C is the maximum number t of errors that C can skillfully decode. All vectors of ${\left(F_{q}\right)}^{n}$ that are orthogonal to every codeword of C consist of the dual code $C^{⊥}$ which is an [n,n−k]-code.*

**Definition** **2**(Hamming Weight)**.**
*The Hamming weight w(x) of a vector x in ${\left(F_{q}\right)}^{n}$ is the number of non-zero entries of x.*

**Definition** **3**(Generator Matrix)**.**
*A generator matrix G of C is the rows that are a basis of C. G is also a k×n matrix.*

**Definition** **4**(Parity-Check Matrix)**.**
*A parity-check matrix H for a linear code C is an (n−k)×n matrix which is a generator matrix for its dual code $C^{⊥}$.*

### 2.2. Coset Decoding

**Definition** **5.**
*Let C be an [n,k]-code over $F_{q}$ and u be any vector in ${\left(F_{q}\right)}^{n}$. The coset of C is defined as follows.*

(1)
u+C={u+x|x∈C}.



**Theorem** **1**(Lagrange)**.**
*Suppose C is an [n,k]-code over $F_{q}$. Then,*
*(i)* 
*Every vector of ${\left(F_{q}\right)}^{n}$ is in some coset of C;*
*(ii)* 
*Every coset contains exactly $q^{k}$ vectors;*
*(iii)* 
*Two cosets either are disjointed or coincided;*
*(iv)* 
*C contains exactly $q^{n-k}$ cosets.*



**Definition** **6**(Coset Leader)**.**
*The coset leader is the vector having a minimum weight in a coset. If a coset contains more than one vector which has the minimum weight, then it is chosen at random as the coset leader.*

**Definition** **7**(Syndrome Decoding)**.**
*Consider H is a parity-check matrix of an [n,k]-code C. In this case,*
(2)$$S\left(y\right)=yH^{T}$$
*is called the syndrome of y, where y is any vector of ${\left(F_{q}\right)}^{n}$, the 1×(n−k) row vector. Moreover,*
(3)S(y)=0⟹y∈C.

**Lemma** **1.**
*Two vectors u and v are in the same coset of C if and only if they have the same syndrome.*


**Corollary** **1.**
*There is a one-to-one correspondence between cosets and syndromes.*


### 2.3. Public-Key Cryptosystems

A cryptosystem is an application of cryptographic methods and ensures the information security services. The cryptosystems can be examined under two titles as the public-key and private-key. Each person has a pair of keys; one is the public-key, and the other is the private-key. The public-key is accessible to the other users; however, the private-key should be stored so that only the owner can access it. Any person can send an encrypted message using the public-key, but only the private-key, which is a pair of public-keys, can decrypt the encrypted message. There is always the mathematical relationship between the public-key and private-key in the public-key cryptosystems. The hardness of two mathematical problems, as integer factoring and discrete logarithm, are used to generate these keys. So, it is impossible to obtain the private-key using the public-key.

The Diffie–Hellman cryptosystem [3] and RSA cryptosystem [4] are pioneers of public-key cryptosystems. However, McEliece [7] and Niederreiter [8] are the first founders of the code-based public-key cryptosystems.

### 2.4. McEliece’s Public-Key Cryptosystem

McEliece’s public-key cryptosystem is the first system based on the algebraic block codes; it was presented in 1978 [7]. In order to construct his cryptosystem, it used a binary (n,k,2t+1) Goppa code *C*. It is clear that *n* is the code length, *k* is the code dimension, and *t* is the error-correcting capacity of *C*. The encryption and decryption algorithms are as follows.

**Private-key:**G,S,P; where *G* is a k×n generator matrix, *S* is any k×k non-singular matrix, and *P* is any n×n permutation matrix.

**Public-key:**$G^{\prime }=SGP$ and *t*.

**Plaintexts:***k* bit vectors *m* over $F_{2}$.

**Encryption:**(4)$$c=mG^{\prime }+e,$$
where *e* is an *n*-bit error vector with Hamming weight *t*. So, *c* is the *n*-bit ciphertext.

**Decryption:**(5)$$cP^{-1}=\left(mS\right)G+eP^{-1}$$
since
(6)c=mSGP.

It is used as the fast decoding algorithm for *C* to correct the error $eP^{-1}$; then, it is found mS and therefore *m*.

### 2.5. Niederreiter’s Public-Key Cryptosystem

Niederreiter [8] proposed a knapsack-type public-key cryptosystem which is based on (n,k,2t+1) linear code *C* over $F_{q}$.

**Private-key:**H,M, and *P*, where *H* is an (n−k)×n parity-check matrix of *C*, *M* is any (n−k)×(n−k) non-singular matrix, and *P* is any n×n permutation matrix, all over $F_{q}$.

**Public-key:**$H^{\prime }=MHP$ and *t*.

**Plaintexts:***n*-dimensional vectors *m* over $F_{q}$ with weight *t*.

**Encryption:**$c=mH^{{}^{\prime }T}$, *c* is the ciphertext of dimension n−k.

**Decryption:**(7)$$c{\left(M^{T}\right)}^{-1}=\left(mP^{T}\right)H^{T}$$
since
(8)$$c=m{\left(MHP\right)}^{T}.$$

It is used as the fast decoding algorithm for *C* to obtain $mP^{T}$ and *m*.

## 3. The System

The construction of our public-key cryptosystem is based on [n,k,2t+1]-code over $F_{q}$. The syndrome-decoding procedure is used for decryption. The public-key and private-key are constructed by each user as follows.

(1)Select a generator k×n matrix *G* of a linear [n,k,2t+1]-code *C* over $F_{q}$, where *t* is the error-correcting capability.(2)Construct a parity-check (n−k)×n matrix *H* from *G* for the code *C*.(3)Select any non-zero syndrom vector *h* which has weight *t* and dimension (n−k).(4)Select a random non-singular (n−k)×(n−k) matrix *M* over $F_{q}$.(5)Calculate n×(n−k) matrix $H^{\prime }=H^{T}\cdot M$, where $H^{T}$ is denoted by the transpose of *H*.(6)The public-key is $\left(H^{\prime },h\right)$.(7)The private-key is (G,H,M).


**Encryption:**


Message: *n* dimension vector *m* over $F_{q}$ with weight *t*.

Cryptogram: $c=mH^{\prime }+h$


**Decryption:**
(1)Calculate $c^{\prime }=cM^{-1}$;(2)Obtain *m* by syndrome decoding $c^{\prime }$ in the code *C*.


Decryption is correct, since
(9)$$w\left(hM^{-1}\right)=w\left(h\right),$$
it can be computed
(10)$$c^{\prime }=cM^{-1}=\left(mH^{\prime }+h\right)M^{-1}=mH^{\prime }M^{-1}+hM^{-1}$$
(11)$$cM^{-1}=mH^{T}MM^{-1}+hM^{-1}$$
(12)$$cM^{-1}=mH^{T}+hM^{-1}$$
(13)$$cM^{-1}-hM^{-1}=mH^{T}$$
and the procedure of syndrome decoding may be effectively used.

**Example** **1.**
*Consider an [4,2,3]-code C over $F_{3}$. The generator matrix G and parity-check matrix H are*

(14)
$$G=\left(\begin{array}{cccc} 1 & 0 & 2 & 2\\ 0 & 1 & 2 & 1 \end{array}\right),$$


(15)
$$H=\left(\begin{array}{cccc} 1 & 1 & 1 & 0\\ 1 & 2 & 0 & 1 \end{array}\right).$$



C={0000,0121,0212,1022,1110,1201,2011,2102,2220}. Select any non-singular matrix $M=\left(\begin{array}{cc} 1 & 2\\ 2 & 0 \end{array}\right).$ The syndromes and coset leaders of C are as follows.
SyndromesCoset Leaders(00) (0000)(11) (1000)(12) (0100)(10) (0010)(01) (0001)(22) (2000)(21) (0200)(20) (0020)(02) (0002)

The size of different cosets of *C* is
$$3^{4-2}=3^{2}=9.$$

So, there are also nine syndrome vectors, which are {00,11,12,10,01,22,21,20,02}. Calculate the matrix
(16)$$H^{\prime }=H^{T}\cdot M=\left(\begin{array}{cc} 1 & 1\\ 1 & 2\\ 1 & 0\\ 0 & 1 \end{array}\right)\cdot \left(\begin{array}{cc} 1 & 2\\ 2 & 0 \end{array}\right)=\left(\begin{array}{cc} 0 & 2\\ 2 & 2\\ 1 & 2\\ 2 & 0 \end{array}\right)$$
and
(17)$$M^{-1}=\left(\begin{array}{cc} 0 & 2\\ 2 & 2 \end{array}\right).$$

Let *h* be the syndrome vector (20). Since d=3, *C* is the corrected t=1 error. So, the public-key is
(18)$$H^{\prime }=\left(\begin{array}{cc} 1 & 1\\ 1 & 2\\ 1 & 0\\ 0 & 1 \end{array}\right),h=\left(20\right)$$
and the private-key is
(19)$$\left(G,H,M\right)=\left(\left(\begin{array}{cccc} 1 & 0 & 2 & 2\\ 0 & 1 & 2 & 1 \end{array}\right)\right),\left(\begin{array}{cccc} 1 & 1 & 1 & 0\\ 1 & 2 & 0 & 1 \end{array}\right),\left(\begin{array}{cc} 1 & 2\\ 2 & 0 \end{array}\right)).$$

**Encryption:** Let the message vector be m=(1000) and h=(20). The cryptogram is
(20)$$c=mH^{\prime }+h=\left(1000\right)\cdot \left(\begin{array}{cc} 0 & 2\\ 2 & 2\\ 1 & 2\\ 2 & 0 \end{array}\right)+\left(20\right)=\left(02\right)+\left(20\right)=\left(22\right).$$

**Decryption:** Calculate
(21)$$c^{\prime }=cM^{-1}=\left(22\right)\cdot \left(\begin{array}{cc} 0 & 2\\ 2 & 2 \end{array}\right)=\left(12\right).$$

Since
(22)$$c=mH^{\prime }+h$$
and
(23)$$H^{\prime }=H^{T}\cdot M,$$
$c^{\prime }$ is also equal to
(24)$$c^{\prime }=\left(mH^{\prime }+h\right)M^{-1}=mH^{\prime }M^{-1}+hM^{-1}=mH^{T}MM^{-1}+hM^{-1}$$
(25)$$c^{\prime }=mH^{T}+hM^{-1}.$$

So,
(26)$$\left(12\right)=\left(m_{1}m_{2}m_{3}m_{4}\right)\cdot \left(\begin{array}{cc} 1 & 1\\ 1 & 2\\ 1 & 0\\ 0 & 1 \end{array}\right)+\left(20\right)\cdot \left(\begin{array}{cc} 0 & 2\\ 2 & 2 \end{array}\right).$$
(27)$$\left(12\right)=\left(m_{1}+m_{2}+m_{3},m_{1}+2m_{2}+m_{4}\right)+\left(01\right)$$
(28)$$\left(12\right)-\left(01\right)=\left(m_{1}+m_{2}+m_{3},m_{1}+2m_{2}+m_{4}\right)$$
(29)$$\left(11\right)=\left(m_{1}+m_{2}+m_{3},m_{1}+2m_{2}+m_{4}\right).$$

We get the message m=(1000) by solving the linear system.

**Proposition** **1.**
*The size of the plaintext is logqnt.*


**Proof.** The plaintext is an n−q tuple word of weight *t*. These are the integers between 1 and nt to the set of words of weight *t* and length *n*. Therefore, the size of the plaintext is logqnt. □

**Proposition** **2.**
*The size of the ciphertext is (n−k).*


**Proof.** Since the ciphertext is a (n−k)−q tuple word, the proof is clear. □

**Corollary** **2.**
*The transmission rate of the new system is*

logqnt(n−k).



**Proof.** The proportion of the number of information symbols to the number of transmitted symbols gives the transmission rate. So, it is
logqnt(n−k).□

**Proposition** **3.**
*Given a syndrome vector y of weight w, the number of eligible h’s is wt(q−1)t.*


**Proof.** It is known that the weight of *h* is *t*, and *h* is non-zero. Thus, the number of non-zero vectors of weight *t* among the vectors of *w* is wt(q−1)t. □

**Example** **2.**
*Let C be the extended binary Hamming code of parameters [8,4,4]. Its packing radius is 1. We examine some properties of the public-key cryptosystem based on C. The size of the plaintext is*

log281=log28=3.


*The size of the ciphertext is*

8−4=4.


*The transmission rate is*

log281(8−4)=0,75.



## 4. Security Comments

In this section, we examine the security of the new system. We recommend using a linear [n,k,2t+1]-code over $F_{q}$. The decryption method is based on the bounded distance decoding task. In order to be a secure public-key cryptosystem, the following conditions should be implemented.

The size of the public-key should be fairly small. In our cryptosystem, this size is (n−k), which is reasonably small.The encryption, decryption, and key generation should be effective. It is computationally simple to create the public-key and private-key. Thus, the encryption and decryption algorithms are too efficient.It should be impossible to reach the plaintext by an attacker.The system should be resistant to all possible attacks. Now, we discuss these attacks for the new system.

### 4.1. Algebraic Attack

The security of a public-key cryptosystem depends on the security of the private-key. So, the first attack will be factorization $H^{\prime }$ to find the private-keys G,H, and *M*. If the code parameters n,k,d are large enough, this attack is impracticable, because it is difficult to recover the factors of $H^{\prime }$. This means the security is ensured with the private-key. The security of the new system is also based on decoding in the code $H^{\prime }$, while $H^{\prime }$ is not only non-equivalent to the code *H* in the cryptosystem, but after multiplying $H^{T}$ by *M* from the right, the error-correction capability of public-key $H^{\prime }$ is unknown. Furthermore, the vector *h* is secret. Thus, the best attack may not carry out the complete decoding.

### 4.2. Generic Attack

The second attack is to reach *m* from *c* without using the private-key. The plaintext is an *n*-*q* tuple word of weight *t*. We require an useful algorithm that matchs the integers between 1 and nt to the set of words of weight *t* and length *n* and vice versa, since the plaintext is a *n*-*q* tuple word of weight *t*. In this case, the attacker will try to repeatedly select *n* bits at random from an (n−k)-bit ciphertext vector and guess *m* based on the *n* selected bits, which is impossible. So, our cryptosystem is strong to all possible attacks. At the same time, the described system presents a general access, which is not for the specific cryptosystem.

Moreover, the probability of no error in the constructing of this system is
$${\left(1-\frac{t}{n-k}\right)}^{n}.$$

Consider the Goppa code, which has the parameters
n=1024,k=524,t=50.

In the public-key cryptosystem constructing based on this code, the probability of no error is
$${\left(1-\frac{50}{500}\right)}^{1024}={\left(0,9\right)}^{1024}.$$

It is a very small number.

## 5. Comparison with the Other Public-Key Cryptosystems

In this section, we compare our system with the other code-based cryptosystem for an [n,k,d]-code *C* over $F_{q}$, where d≥2t+1. We denote by S,R,T, and *K*, respectively, the size of plaintext, ciphertext, the transmission rate, and the dimension of the public-key.

The new system is a further development of the McEliece and Niederreiter cryptosystems. McEliece’s system is constructed based on binary linear codes, but both Niederreiter’s and our new system are constructed based on linear codes over $F_{q}$. Especially, we use the bounded distance decoding to construct our system. In the new system, as the public-key is smaller than McEliece’s cryptosystem, it is more useful in industry. Moreover, as it is seen in Table 1, the plaintext is a word of small weight, which is one of the coset leaders, and the number of operations involved during the encryption is less than McEliece’s cryptosystem. Furthermore, it is seen that the public-keys in our system and Niederreiter’s system are equivalent. However, our system is more effective than Niederreiter’s cryptosystem, since it is easier to generate the pieces of keys. This condition increases the security. It is impossible to reach the private-key with public-key by an attacker in the new system. In addition, the plaintext cannot be calculated even if the public-key and ciphertext are known by an enemy cryptanalyst. When the transmission rates of systems are compared, it is noticed that the proposed system has the bigger magnitude. That is, the encryption is faster than the others. So, it is more reliable by means of security.

## 6. Conclusions

We presented a new public-key cryptosystem based on error-correcting codes in this study. This system refers to the class of cryptosystems based on the bounded distance decoding task. The sizes of the plaintext and ciphertext of the system are calculated. Therefore, the transmission rate is given. The possible attacks are considered. It is determined that the new system stands well when compared with known systems.

## Figures and Tables

**Table 1 entropy-24-00498-t001:** Comparison with other schemes.

System	[7]	[8]	This Paper
*S*	*k*	*n*	*n*
*R*	*k*	n−k	n−k
*T*	$\frac{k}{n}$	log2nt(n−k)	logqnt(n−k)
*K*	*k*	n−k	n−k

## Data Availability

Not applicable.

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
