# Peer review of "Public-Key Cryptosystems and Bounded Distance Decoding of Linear Codes"

_entropy, 2022, doi:10.3390/e24040498_

Round 1

Reviewer 1 Report

This paper proposes a public-key cryptosystem based on the error correcting codes. 

Pros: 

The paper gives a concrete example to justify their construction.

Cons:

The problem has not been well motivated in the Introduction. It is unclear why we need another public-key cryptosystem based on error-correcting codes. It is also unclear why such a design could be challenging.

The paper should provide a little more formal security analysis to justify that the proposed design is secure.

What are the advantages of the design proposed in this paper compared to that in [14]. It does not seem it can achieve something much better than [14].

The presentation needs to be improved:

1) There are too many paragraphs in the Introduction section. Please combine them into few paragraphs.

2) Each table should have a title and id. The table in Sec. 5 seems to have missed both title and id.

3) There are a lot of grammar errors. For example, "Error correcting codes form an important topic in Information theory", "It is seen that too small number.", etc.

Author Response

Dear Reviewer,

Thanks for your valuable comments. My response is as follows.

Best Regards

Selda ÇALKAVUR

  • The introduction is improved. In this context, it is explained why need another public-key cryptosystem based on error-correcting codes and why such a design is difficult.
  • The security comments (Section 4) are improved. It is explained why the system is more secure.
  • The Section 5 is improved. It is especially compared with Niederreiter’s Cryptosystem in detail.
  • Some paragraphs are combined in the Introduction.
  • The table is numbered and named in Section 5.
  • The grammar errors are corrected.

Reviewer 2 Report

The paper addresses an interesting issue of design code-based public key crypto system, but the proposal should be justified much more formally and much more in details. 

When a cryptographic system is proposing, cryptographic security evaluation is the most important issue. The given analysis in section 4 is far from appropriate one, and more formal security evaluation should be performed, at least following the guidelines given in: Jonathan Katz, Yehuda Lindell: Introduction to Modern Cryptography, 3rd Edition,  ISBN 9780815354369, Chapman and Hall/CRC, 648 Pages, Dec. 2020.  

Security evaluation and bounded distance decoding should be considered in more details, maybe using the approach reported in the following reference:  Zhe Li, San Ling, Chaoping Xing, Sze Ling Ye, “On the Bounded Distance Decoding Problem for Lattices Constructed and Their Cryptographic Applications”, IEEE Trans. Information Theory, Vol.: 66, Issue: 4,  pp. 2588 - 2598    April 2020.

Instead of small toy example coding schemes the more realistic should be considered.

Final merit of the results reported should be reconsidered after a major revision the current manuscript.

Author Response

Dear Reviewer,

Thanks for your valuablae comments. My response is as follows.

Best Regards

Selda ÇALKAVUR

  • The introduction is improved. In this context, it is explained why need another public-key cryptosystem based on error-correcting codes and why such a design is difficult.
  • The security comments (Section 4) are improved. It is explained why the system is more secure. It is used the reference“ Zhe Li, San Ling, Chaoping Xing, Sze Ling Ye, “On the Bounded Distance Decoding Problem for Lattices Constructed and Their Cryptographic Applications”, IEEE Trans. Information Theory, Vol.: 66, Issue: 4,  pp. 2588 - 2598    April 2020.”
  • The Section 5 is improved. It is especially compared with Niederreiter’s Cryptosystem in detail.
  • It should not be considered that the small toy example. We have to explain with small parameters to understand our system in the cryptographic papers. Because the examples with big parameters cover a very large field in the paper. We already mention that it is benefical to use with big parameters for the security in the study.
  • The grammar errors are corrected.

Round 2

Reviewer 1 Report

Please better organize the differences between the proposed design and Niederreiter’s cryptosystem. You can itemize the differences in the Introduction section. Also, make sure the differences can be justified in Section 5 in which you compare the proposed design with others.

Author Response

Dear Reviewer,

My response is as follows.

Best Regards

Selda ÇALKAVUR

It is itemized the differences between the proposed design and Niederreiter’s system in the Introduction section.
2) The section 5 is improved.
3) The grammar errors are corrected.
